# The Dawn of the Age of Multi-Parent MAGIC Populations in Plant Breeding: Novel Powerful Next-Generation Resources for Genetic Analysis and Selection of Recombinant Elite Material

**DOI:** 10.3390/biology9080229

**Published:** 2020-08-16

**Authors:** Andrea Arrones, Santiago Vilanova, Mariola Plazas, Giulio Mangino, Laura Pascual, María José Díez, Jaime Prohens, Pietro Gramazio

**Affiliations:** 1Instituto de Conservacióny Mejora de la Agrodiversidad Valenciana, Universitat Politècnica de València, Camino de Vera 14, 46022 Valencia, Spain; anarol1@etsiamn.upv.es (A.A.); maplaav@btc.upv.es (M.P.); giuman2@upvnet.upv.es (G.M.); mdiezni@btc.upv.es (M.J.D.); jprohens@btc.upv.es (J.P.); 2Department of Biotechnology-Plant Biology, School of Agricultural, Food and Biosystems Engineering, Universidad Politécnica de Madrid, 28040 Madrid, Spain; laura.pascual@upm.es; 3Faculty of Life and Environmental Sciences, University of Tsukuba, 1-1-1 Tennodai, Tsukuba 305-8572, Japan

**Keywords:** MAGIC population, mapping populations, RILs, QTLs, association analysis, breeding resources, public–private partnerships, climate change

## Abstract

The compelling need to increase global agricultural production requires new breeding approaches that facilitate exploiting the diversity available in the plant genetic resources. Multi-parent advanced generation inter-cross (MAGIC) populations are large sets of recombinant inbred lines (RILs) that are a genetic mosaic of multiple founder parents. MAGIC populations display emerging features over experimental bi-parental and germplasm populations in combining significant levels of genetic recombination, a lack of genetic structure, and high genetic and phenotypic diversity. The development of MAGIC populations can be performed using “funnel” or “diallel” cross-designs, which are of great relevance choosing appropriate parents and defining optimal population sizes. Significant advances in specific software development are facilitating the genetic analysis of the complex genetic constitutions of MAGIC populations. Despite the complexity and the resources required in their development, due to their potential and interest for breeding, the number of MAGIC populations available and under development is continuously growing, with 45 MAGIC populations in different crops being reported here. Though cereals are by far the crop group where more MAGIC populations have been developed, MAGIC populations have also started to become available in other crop groups. The results obtained so far demonstrate that MAGIC populations are a very powerful tool for the dissection of complex traits, as well as a resource for the selection of recombinant elite breeding material and cultivars. In addition, some new MAGIC approaches that can make significant contributions to breeding, such as the development of inter-specific MAGIC populations, the development of MAGIC-like populations in crops where pure lines are not available, and the establishment of strategies for the straightforward incorporation of MAGIC materials in breeding pipelines, have barely been explored. The evidence that is already available indicates that MAGIC populations will play a major role in the coming years in allowing for impressive gains in plant breeding for developing new generations of dramatically improved cultivars.

## 1. Introduction

The Green Revolution of the 1960s, together with substantial investment in plant breeding since then, enabled a significant increase in crop yields, especially in staple grain crops like wheat, rice, and maize [1,2]. Despite this, by 2050, agricultural production may need to be increased by 60–110% compared to current levels [3,4]. Though crop yields continue to increase globally, climate change represents a tremendous challenge for achieving this objective. In fact, recent research suggests that some major crop yields have already stagnated or even been reduced by the impact of climate change [5].

The success of conventional breeding has been enhanced by genetic transformation technologies that were applied since the 1980s, allowing for the achievement of important advances in plant breeding [3]. Nevertheless, many of those technologies have been mostly focused on monogenic traits, while many major agronomic traits of interest are quantitative, controlled by multiple loci, and generally have a large environmental influence [6,7,8,9], thus posing a challenge to breeders due to the limited efficiency of breeding based on phenotypic selection.

Geldermann [10] introduced the term quantitative trait locus (QTL) to describe “a region of the genome associated with an effect on a continuous trait.” The identification of QTLs that significantly contribute to improve relevant yield and quality traits is a key factor in promoting a new Green Revolution [11]. By using marker-assisted selection (MAS), introgression breeding and the pyramiding of QTLs can be efficiently achieved to obtain dramatically improved cultivars [12,13]. However, the identification of QTLs underlying quantitative traits remains a challenge for plant breeders [14]. Four basic elements are required to detect QTLs in crops: (i) an appropriate segregating population with a high genetic variability and contrasting parents for the target phenotype; (ii) marker systems that allow for the genotyping of the population; (iii) reproducible quantitative phenotyping methodologies; and (iv) an appropriate experimental design to evaluate the environmental effects and statistical methods to detect and locate QTLs [7,15,16,17,18].

The use of experimental and germplasm populations has been of great relevance for mapping QTLs related to agronomic traits of interest [19]. A new singular type of multiparent population, the so-called multi-parent advanced generation inter-cross (MAGIC), was firstly proposed by Mackay and Powell [20]. MAGIC populations are a set of immortal fixed lines with a genome that is an admixture of the genomes of multiple founder parents [16,20,21]. Already available theoretical and experimental studies on MAGIC populations have revealed that they can make a relevant contribution for a dramatic improvement in detection of QTLs and the development of elite breeding material, as well as to a dramatic improvement in genetic advances in plant breeding [22,23].

Though some recent reviews on multi-parent populations have been published due to the growing interest of MAGIC populations [24,25], our focal point is on the potential of MAGIC populations for plant breeding, including comprehensive information on the different types of mapping populations for QTL detection and the advantages and limitations of MAGIC populations, as well as strategies for development and available software for their genetic analysis. We also provide an overview of MAGIC populations already developed in plants and their utility for the detection of QTLs and breeding. In addition, we propose new approximations and prospects opening new avenues for MAGIC population expansion to inter-specific or MAGIC-like populations, as well as their incorporation into breeding pipelines.

## 2. Overview of Experimental Populations and Germplasm Collection for Traits Dissection

### 2.1. Bi-Parental Populations

Traditionally, genetic linkage mapping studies for QTL detection in plants have used data collected from experimental populations, especially from bi-parental populations, like F2 and first backcross (BC1) generations [26]. They can be directly analyzed [27,28] or studied after fixing by selfing until homozygous immortal populations [29], such as recombinant inbred lines (RILs), backcross inbred lines (BILs), near isogenic lines (NILs), or doubled haploid populations (DHs), are obtained [15,21] (Figure 1). These methodologies have been used in many genetic studies to identify QTLs and to clone major genes underlying QTLs linked to key traits in many crops [7,29,30,31].

In biparental populations, two inbred lines are usually crossed to produce one or more segregating progenies [26]. Parents are selected based on their genetic and phenotypic diversity for a trait of interest, allowing for the detection of genomic regions associated with the target trait by the reconstruction of progeny genomes from the founder haplotypes [32]. However, populations derived from bi-parental crosses only capture a small picture of the genetic factors that affect target traits in the species and suffer from a shortage of diversity due to the narrow genetic base that is limited to both parents [17,32,33]. In addition, the limited opportunities for genetic recombination events, especially in F2, BC1, and DH populations, limits the resolution for QTL detection [34,35]. In these approaches, only two alleles are generally analyzed, with a maximum resolution of 10–30 cm, as just a single opportunity of recombination has been possible during F1 meiosis [36]. As a result, loci are mapped with low resolution and with large genetic support intervals [33,35]. The same happens even in RILs because the number of efficient recombination decreases in advanced generations, as the proportion of the genome in heterozygosis is reduced to a half in each generation of selfing [37]. Furthermore, the QTLs detected in a biparental population might not be expressed in other genetic backgrounds [17]. 

### 2.2. Germplasm Populations and Germplasm Collections

The continuous reduction of high-throughput genotyping costs together with the development of new genomic technologies has led to an emergence of QTL mapping resources, alternative to biparental populations, that demand an increase in marker density [38,39,40]. Genome-wide association studies (GWAS) have become one of the main genomic tools to dissect QTLs and genes underlying complex traits [40,41]. Association mapping takes advantage of high-genetic-diversity panels, including collections of selected individuals with unknown kinship and the historical accumulation of recombination events during thousands of generations [42] (Figure 2). The application of GWAS for the more precise mapping of novel genes related to complex agronomic traits in crop plants has been demonstrated in many studies [43,44]. In spite of that, the main limitations of GWAS are linkage disequilibrium (LD) (which may vary greatly among genome regions with different block length along chromosomes), population sub-structure (which can lead to inaccurate results), and unbalanced allele frequencies [41,45]. Also, GWAS requires very large samples and many markers to have enough power to detect genomic regions of interest and its efficiency is limited by unknown pedigrees and missing parental information [17,32]. Furthermore, they may be suboptimal to identify rare alleles conferring interesting phenotypes [46]. 

### 2.3. Multi-Parent Populations 

Alternatively, multi-parent populations may offer solutions to the main drawbacks of bi-parental and germplasm populations [9,20,24,37]. Multi-parent populations or multi-parent cross designs (MpCD) have been developed in many crop species throughout the history of scientific plant breeding [32,47,48]. The discovery of genetic male sterility (GMS) systems in the 1960s facilitated their adaptation to crops where artificial hybridization is challenging [49,50]. 

Multi-parent populations are produced by crossing more than two inbred founder lines, and they represent a bridge between linkage mapping (traditional bi-parental crosses) and association mapping (GWAS) approaches. This new approach dramatically increased mapping resolution by incorporating multiple founders with increased genetic and phenotypic diversity. Thanks to the increasing development of more sophisticated tools in high-throughput genomic technologies and statistical analysis power, multi-parent populations can be employed in many genetic mapping studies [20]. 

There are many types of MpCD, but here, we focus on MAGIC populations [16]. MAGIC populations are panels of RILs that are fine-scale mosaics with theoretically equal proportions of the founder genomes. Therefore, MAGIC populations occupy an intermediate standing and represent a compromise between the much greater complexity found in naturally occurring accessions and the extreme simplicity of a diallelic system of RILs [51]. In this way, MAGIC populations are considered an emerging and powerful next-generation mapping resource within plant genetics, combining high-genetic recombination and diversity to dissect the structure of complex traits for improving breeding programs [21,24,38,40,52]. Despite being very similar to other multiparent populations like the heterogeneous stock and Collaborative Cross populations used in mouse genetics, the MAGIC populations approach was first proposed for being used in crops in 2007 [16,20].

Multiple founders in MAGIC populations enrich populations with higher allelic diversity when compared to those derived from typical bi-parental crosses, whereas multiple cycles of parental inter-crossing result in a set of rearranged genomes with a high level of fragmentation, thus giving greater opportunities for recombination and dramatically increasing the power of QTL detection [35,40,53]. That makes them fit for both gene mapping and the effective generation of pre-breeding material. In this way, MAGIC populations provide an ideal platform and a common route for the discovery and high-resolution dissection of the QTLs and genes responsible for complex agronomic traits for crop improvement [16,20].

MAGIC populations have been set up in many model species, thus demonstrating their power to detect polymorphisms underlying QTLs or genes related to complex traits of interest [37]. Currently, MAGIC and MAGIC-like populations are available in a wide range of crop species including cereals, legumes, vegetables, fruit trees, and industrial crops, while many others are under development. Due to the wide genetic base of MAGIC populations, they are useful for QTL and gene discovery, the enhancement of breeding populations, and the development and release of new varieties [52]. In this way, they facilitate investigation, not only in the genome architecture but also in its relationship with phenotypic traits and environment effects [17].

## 3. Advantages and Limitations of MAGIC Populations

### 3.1. Advantages 

MAGIC populations constitute a useful resource for genetic studies to the scientific community and, as indicated before, present clear advantages compared to classical bi-parental and germplasm populations. Thanks to their potential, they could be basically used for two different non-excluding purposes: (i) permanent immortal mapping populations for precise QTL locating and (ii) the development of new elite material to be directly released or to be included in breeding pipelines as pre-breeding materials [35]. 

Regarding QTL detection, MAGIC populations are very useful genetic materials for linkage and association methodologies. The large number of accumulated recombinant events achieved over multiple rounds of inter-crossing and selfing improves QTL mapping accuracy, thus creating an opportunity to identify gene-trait associations with a greater resolution [20,24,33,37] (Table 1). The increased recombination rate also facilitates further reductions in LD and provides little or no population structure that could lead to false-positive results [16,20,33,34,54]. 

The combination of multiple founders provides a higher genetic and phenotypic diversity within a single mapping population, thus increasing the number of QTLs that segregate in the cross and promoting novel rearrangements of alleles [35,51] (Table 1). An optimal founder selection allows for the targeting of multiple traits in a mixed background of parental genomes [35,37]. For instance, a QTL may not express in a single background of one founder but may be expressed in the MAGIC population mixture. MAGIC populations provide the opportunity to simultaneously assess the effect of multiple alleles at a locus and to study the interactions of cytoplasm effects, genome introgressions, and chromosomal recombination involving allelic diversity across the genome [16,33,35]. Their potential for putative causal polymorphism identification underlying the QTLs, when coupled with available genome sequences, can be further enhanced to exploit current rapid advancements by the application of emerging high-throughput genotyping and phenomics platforms in the post-genomics era [37]. These future ground-breaking approaches allow for the study of QTL functions and interactions across multiple traits within MAGIC populations [38,39]. 

Regarding the potential use of the genetically diverse and immortal RILs that constitute the final MAGIC population, they provide a promising and useful germplasm to be exploited in breeding programs for the development of improved lines and hybrids or to be released as new cultivars [35,40]. They can also be analyzed across a wide range of environments to increase the understanding of gene–environment interactions (G×E) and phenotypic plasticity, providing a permanent resource to study the basis of phenotypic traits [24,37,55]. 

MAGIC populations represent an important pre-breeding resource, not only for the final lines already obtained but also for the possibility of developing new combinations. Previously identified QTLs and the novel ones detected in the final MAGIC could be synergically exploited to select the best MAGIC RILs as super trait-donor lines. In this way, they would serve as an advanced source for breeding programs to pyramidize novel combinations of QTLs. This strategy would be similar to the multi-parent advanced generation recurrent selection (MAGReS) approach proposed by the authors of [17] but with some important differences. The MAGIC RIL sets selected for inter-crosses are more advanced and nearly homozygous (ideally F6–F8), and their selection can be targeted by using prior knowledge of bi-parental QTLs together with QTL haplotypes and predicted breeding values based on genome-background diversity [52]. The resulting MAGReS lines will be fixed for positive haplotypes and will carry novel QTL combinations in other unknown loci conferring different unexpected traits. Therefore, selected MAGIC lines can be a valuable resource for genetic improvement as super trait-donors that contribute to new QTL combinations [52].

### 3.2. Limitations

Though MAGIC populations overcome the main limitations of bi-parental and germplasm populations, they have some disadvantages linked to their development (Table 1). First and foremost, they need more investment in time and greater efforts to be developed due to their convoluted crossing scheme involving multiple founder lines and a large number of selfing generations to obtain an almost homozygous final MAGIC population [37,38]. The crossing of selected parent lines and the following generations of inbreeding take several years due to the limitation of one-to-two generations per year of most of the crops, thus slowing down the development of the MAGIC population. The use of a DH population is the shortest way to produce fully homozygous populations, but they will incorporate recombination events that are only produced during the initial and advanced stages of inter-crossing. In addition, for many crops, this technique is strongly species- and genotype-dependent, requiring a prior establishment of an optimized and efficient protocol that is usually very laborious [56]. Another option is the so called “speed breeding” approach, which has been recently implemented in some cereal and model legume crops. “Speed breeding” greatly shortens generation time and accelerates breeding and research programs by growing plant populations under controlled photoperiod and temperature regimes [57,58]. Controlled environments accelerate the development rate of plants and the harvesting and germination of immature seeds, thereby reducing generation time and increasing the number of plant generations obtained in one year. However, the facility costs required for controlled-environment growth chambers and the use of continuous supplemental lighting can make the cost of a project soar (Table 1). MAGIC populations that are highly powerful for fine genetic mapping also require a larger population size compared to a bi-parent population, which not only hinders population development but also limits phenotypic traits to evaluate. Therefore, population goals should be clearly defined before initiating population development since, for example, the identification of epistatic interactions requires a larger population size.

The presence of multiple founders involved in population development makes the process logistically challenging and labor-intensive [40]. Many molecular markers are required for the analysis of MAGIC populations due to their complex genomic structure conformation because of the multiple parental admixture [17] (Table 1). Knowledge of the population haplotype structure offers a more informative estimation of QTLs; however, due to the number of founder genomes, the direct observation of each parental contribution to a QTL remains a challenge [59,60]. The careful selection of founders is one of the most important design considerations because it ensures that the MAGIC population is going to be relevant as a long-term genetic resource panel [17,24]. When wild species are in the panel of founder parents to broaden the genetic diversity, genetic and genomic incompatibility may appear. This is a handicap that may lead to a large reduction in the number of progenies derived from a specific MAGIC design. Usually, greater progeny reductions occur in early generations, since detrimental genes or combinations of such are manifested at the beginning, and these deficient lines leave no offspring. The subsequent occurrence of natural or artificial (either inadvertent or intentional) selection in the following generations is also extremely for the development of MAGIC populations. If genetic bottlenecks from a great reduction of lines occur in any generation, final lines will have family relationship that reduces the genetic diversity and creates population structure [17]. In addition, it is important to consider that the complexity of the funnel crossing schemes poses a potential intermating bias that can result in assortative mating instead of the assumed random mating [40]. 

There are different MAGIC schemes including, among other approaches, four-, eight-, or 16-way recombinant inbred lines obtained by selfing [61]. With a higher number of founders, intermating becomes more and more unmanageable due to the number of individuals in each generation needed to achieve a complete admixture. Without the complete control of the reproduction, this complexity increases the chances of mating between individuals with similar genotypes, thus causing assortative mating and a deviation from random mating expectation [62]. Assortative mating introduces a generation of genetic subgroupings and distorts LD, causing wrong or spurious associations [63,64,65,66]. 

When focusing on locations and mapping QTL frequency, bi-parental RILs regularly display ordinary or bi-modal distributions, whereas MAGIC populations show a continuous distribution that tends to deviate from their founders due to the unequal contribution of each parental genome in individual MAGIC lines [67]. Broman [68] suggested that the final MAGIC composition will directly depend on the cross-design selection and funnel structure chosen to construct the population. For this reason, before embarking on a MAGIC population development, it is important to make premeditated decisions on the purpose of the MAGIC population to be developed.

## 4. MAGIC Development Strategies

### 4.1. Cross-Designs

MAGIC populations can be developed by different breeding designs that give rise to different population structures (Figure 3). Basically, the two main approaches to develop MAGIC populations are the following [69]:

(a) The “funnel” approach, in which the development of the MAGIC population starts with a funnel breeding scheme (Figure 3A). Typically, the funnel is created by several generations of inter-crossing among a number (n) of elite parental lines to obtain n/2 F1 hybrids, which are subsequently intercrossed in a set mating design to combine the genomes of all founders in the progeny lines.

(b) The “diallel” approach, in which the parents are crossed according to a diallel or half-diallel design, thus inter-crossing the parents in multiple funnels by a half-diallel mating system (two-way crosses), followed by the inter-crossing of the resulting F1s until all the founders are represented in a single generation (Figure 3B). The diallel or half-diallel mating design is similar to the funnel approach, but it covers all possible two-way, four-way, and eight-way cross combinations. In the case of eight parental lines, a total of 28 two-way, 14 four-way, and seven eight-way crosses could be performed to develop the MAGIC population. 

The “diallel” approach is the preferred cross-design used in the currently available MAGIC populations because it includes a higher diversity of allele combinations. However, in some cases, such as in tomato MAGIC populations, the “funnel” design has been chosen [37,70]. This could be due to the fact that in the MAGIC population developed by Pascual et al. [37], one of the founders was a variety with a small flower size, which makes hybridization difficult when used as a female parent. On the other hand, in the tomato MAGIC population developed by Campanelli et al. [70], a wild species was used, thus hindering the multiple crossing process. Reducing the number of hybridizations needed by following the funnel design allows for the faster advancement of the development of the MAGIC populations.

Once all founder genomes are admixed, some family inter-crossing can be performed to increase recombination and to finally achieve better mapping and QTL identification resolution, like in the barley and wheat MAGIC populations developed by Sannemann et al. [60] and Stadlmeier et al. [71], respectively. In any case, the final objective of both the funnel and diallel designs is the achievement of individuals with a high degree of homozygosis, which can be obtained by single-seed descent (SSD) or through double haploid production (Figure 3C). For instance, Sannemann et al. [60] produced approximately 5000 barley MAGIC doubled haploid lines after eight-way inter-crossing. Through DH production via anther and microspore culture, they achieved completely homozygous individuals in a single step, significantly shortening the breeding program. Even though the doubled haploid production can be very fast in developing a MAGIC population when an optimal protocol is established, additional rounds of selfing generations after the admixing of genomes via hybridization can introduce novel recombination events, although in a lower ratio than during the initial and advanced stages of inter-crossing [17]. When the development of highly homozygous lines is achieved through self-fertilization after hybridizations, five-to-eight generations of SSD are ideally recommended, resulting in RILs with complex pedigrees and with expected levels of genome heterozygosity between 3.125% and 0.319% for five and eight generations, respectively [17]. As a result, each RIL carries a mosaic of genome blocks theoretically contributed in equal proportions by all founders [17,21,33]. Blind SSD with no intended selection for preferable seed characteristics, plant type, or yield components was found to help create high diversity in morphological and agronomic traits in MAGIC populations [52]. 

Full-sib families obtained after several generations of selfing, which still have a moderate-to-high percentage of heterozygosity, can be used for coarse mapping and to develop NILs for the parts of the genome that still are in heterozygosis [72]. NILs are lines differing in a small genomic region or introgression fragment in an otherwise homogeneous genetic background, and they are usually obtained by repeated backcrossing to a recurrent parent. In this way by deriving NILs in advanced generations of selfing in MAGIC populations, there is no need of backcrossing for the precise location of the gene/QTL responsible for a trait of interest. Final lines could be further used as pre-breeding materials, or they can be directly released and registered as new varieties.

### 4.2. Founder Selection

One of the most important decisions to take prior to the development of MAGIC populations is the selection of founders that are going to give rise to the final population. Ideally, founders should cover a broad phenotypic, genetic, and geographic diversity to exploit the potential of the population and to obtain a highly informative resource panel [17] (Figure 4). Among MAGIC population founders, wild relative species or landraces coming from different origins that are very well adapted to their specific growing conditions could be candidates, in addition to commercial cultivars or elite breeding lines with desirable traits. The founders must be well characterized at the molecular and physiological levels to produce a practical resource. If the founders are also previously resequenced, it enables better design strategies to identify causal polymorphisms underlying QTLs, such as an efficient marker selection [37]. It is important to not only ensure high diversity at the genetic and phenotypic level but also to carefully manage it. The selection of wild relatives to broaden genetic diversity could lead to incompatibilities, linkage drag due to a lack of recombination in some regions of the genome, and the selection of certain phenotypic traits like a lack of seed dormancy or early flowering time. In these cases, the number of offspring families may suffer a reduction that could endanger the development of the MAGIC population [17]. Currently available MAGIC populations have generally been obtained using only intraspecific diversity, as in most cases their founders are landraces, improved and adapted breeding lines, or commercial cultivars. Though they are an interesting option to broaden population diversity, difficulties associated to the selection of wild species as founders make the development of inter-specific populations a challenge. However, some MAGIC populations incorporating one or more wild relative accessions have been developed or are under development [70,73].

An example of an inter-specific population already available is the tomato MAGIC population developed by Campanelli et al. [70]. They selected a *Solanum cheesmaniae* accession as a founder due to a very large interesting dataset of traits such as biotic and abiotic stress tolerance, yield, and resilience [74]. 

### 4.3. Population Sizes

Large population sizes are essential in each generation during the development of the MAGIC population to avoid genetic bottlenecks in some generations but also to have enough power for QTL identification. A mapping population of at least 50–250 individuals is generally required for coarse QTL mapping [15,18]. It is well-known that as population size increases, the power and resolution of QTL mapping also increases [34,61]. In addition, the larger the population size, the weaker the LD within and between chromosome blocks and the lower the number of genetic markers to ‘‘tag’’ a haplotype [45,75,76]. However, larger population sizes also imply higher difficulties to characterize and maintain a population. Certainly, there must be an equilibrium between population size and the efforts invested to establish the population. 

An optimal population size strongly depends on genome size [17,34,52] (Figure 5). Organisms with large genomes require an offspring of at least 500 individuals to provide a resolution power of the sub-centimorgan range, enough to detect singe QTLs that explain 5% of the phenotypic variability [34]. Organisms with smaller genomes can afford to reduce their population size while maintaining the same resolution power. For instance, due to relatively small cowpea diploid genome (620 Mb), 305 MAGIC RILs are enough to cover its whole genome with fine resolution. A comparable level of physical resolution in an organism with a genome in the 5-Gb range (barley or diploid wheat) would require a population size of about 2500 RILs [52]. MAGIC populations in plants with larger genomes should be larger than those with small genomes, but due to practical considerations and limitations, this is not always the case (Figure 5).

Though a desired population size should be established before initiating MAGIC population development, on a practical level, maintaining it during generations is not very easy, and a certain percentage of losses of families during the process may occur, depending on the species and the founders used. In this way, as population progresses, the aimed population size could be affected by the presence of plants with poor development, fruit setting, or by the appearance of parthenocarpic fruits. In addition, if a wild species has been selected as a founder, prior drawbacks may increase. The largest losses of families take place in the early generations as selection for appropriate development, high fertility, and seed production takes place.

## 5. Analysis Software for Genetic Gap Construction and QTL Mapping

The construction of genetic maps for marker–trait association analyses in MAGIC populations is challenging due to their complex cross-design. R/mpMap is one of the few pieces of software available for MAGIC map construction [59]. This R package has been adapted to four-way and eight-way MAGIC populations and has been used in the wheat and tomato MAGIC populations developed by Mackay et al. [38] and Pascual et al. [37], respectively. More recently, R/mpMap2 and magicMap R packages have been extended to other multi-parental populations [77,78]. 

Regarding QTL mapping software, a wide range of statistical analysis tools for bi-parental crosses is available; however, this existing analysis software cannot be directly applied to multiparent inbred line crosses [7,79]. While bi-parental analysis software only includes two allelic segregation patterns, multi-parent software must include all possible patterns according to the number of founders that are involved in the MAGIC population development. The higher complexity of multi-parent population designs requires a flexible and general framework for analyses capable of reconstructing founders’ haplotype mosaics, to impute whole-genome genetic variants, and to handle multiple founder alleles and their population structures [61]. Therefore, different R packages started to appear due to the urgent need for analysis tools that are capable of handling the complexities associated with these advanced populations (Figure 6).

The first available QTL mapping software used for multi-patent population analysis was the R package HAPPY [80], an interval mapping approach method based on founder probabilities. This software was used for the analysis of the *Arabidopsis thaliana* MAGIC, the first MAGIC population developed [51] (Table 2). However, it was quickly replaced by R/qtl [81] and R/mpMap [59]. While R/qtl incorporated different functions to analyze more complex populations, R/mpMap was developed as a comprehensive set of methods for analyzing multiparent designs with a greater flexibility in pedigree definition. Unlike R/qtl or R/HAPPY, R/mpMap was developed to accommodate linear mixed models to simultaneously assess genetic and environmental variation. Due to the clear advantages offered by R/mpMap, it is the most widespread and widely used software for MAGIC population analysis [33,37,52,60,71,77] (Table 2).

However, MAGIC populations can also be directly analyzed for establishing marker–trait associations via GWAS [20,82,83]. The most used alternative is the Trait Analysis by aSSociation, Evolution and Linkage (TASSEL) software that uses phenotypic and genotypic data [84]. For controlling populations and family structures, this software implements general linear model and mixed linear model approaches. This useful software has been widely used for rice, cotton, wheat, sorghum, and *Brassica juncea* MAGIC population analyses [35,40,85,86,87]. Furthermore, some R packages are also available to conduct GWAS such as the so-called Genome Association and Prediction Integrated Tool (GAPIT) that implements advanced statistical methods, including mixed linear models [88]. This genome association and prediction integrated tool has been used to analyze wheat, cotton, and maize MAGIC populations [38,85,89].

A whole-genome average interval mapping (WGAIM) approach was first proposed to accommodate all genotype information in a single model for bi-parental population analysis [90], before being extended to MAGIC populations [91]. This analysis software was used in the wheat MAGIC population developed by Rebetzke et al. [92] (Table 2). It is useful for QTL analysis with multiple alleles in a multi-environment or multi-trait data in a linear mixed model framework, which results in a greater understanding of traits or environments and the relationship between them. For this reason, they renamed this approach to multivariate multiparent (MVMP) WGAIM [91]. 

More recently, R/qtl2 software [61] expanded the scope of the widely used R/qtl software package. This redesigned R package includes implementations of many different multi-parent population cross-designs and it is suited for high-dimensional genotype and phenotype data. Therefore, it includes numerous quality-control assessments as QTL genome scans by using the Haley–Knott regression method [93], linear mixed models to account for population structure, and best linear unbiased prediction (BLUP) based estimates of QTL effects.

**Table 2 biology-09-00229-t002:** Current status of available MAGIC populations in the model species *Arabidopsis thaliana* and in crop plants, including the cross-design, founders, number of RILs in the MAGIC populations, target traits, and software analysis used for QTL detection.

Crop	Design	Founders	Final RILs Population	Target Traits	QTL Analysis Software	Reference
*Model species*						
*A. thaliana*	19-way, diallel	Natural accessions	1026 S6	Germination date and bolting time	HAPPY	Kover et al. [51]
	8-way, diallel	Natural accessions	532 F5	Flowering time and leaf morphology	GenStat	Huang et al. [94]
*Cereals*						
Wheat	4-way, diallel	Cultivars	1579 F6	Plant height and hectoliter weight	mpMap	Huang et al. [33]
	8-way, diallel	Cultivars	–	Plant height and hectoliter weight	mpMap	Huang et al. [33]
	8-way, diallel	Cultivars	1091 F7	Awning	GAPIT	Mackay et al. [38]
	4-way, diallel	Commercial cultivars	1458 F6:7	Coleoptile length and thickness and shoot length	WGAIM	Rebetzke et al. [92]
	60-way, NAM-like	Breeding lines	1000 S4	Flowering time	In-house software	Thépot et al. [95]
	4-way, diallel	Cultivars	>338 F8	Plant height and grain yield	TASSEL	Milner et al. [86]
	8-way, diallel	Cultivars	2125 F4	Plant height	GWAS	Sannemann et al. [83]
	8-way, funnel	Breeding lines	516 F6:8	Powdery mildew resistance	mpMap	Stadlmeier et al. [71]
	8-way, diallel	Elite lines and cultivars	>3000 S2:5	Number of recombination events	mpMap	Shah et al. [77]
Rice	8-way *indica*, diallel	Elite and modern cultivars	1328 S7	Biotic/abiotic stress and grain quality	TASSEL	Bandillo et al. [35]
	8-way *japonica*, diallel	Elite and modern cultivars	500 S5	Biotic/abiotic stress and grain quality	TASSEL	Bandillo et al. [35]
	8-way MAGIC-plus, diallel	Elite and modern cultivars	S4 (in progress)	Biotic/abiotic stress and grain quality	TASSEL	Bandillo et al. [35]
	16-way MAGIC global, diallel	Elite and modern cultivars	–	Biotic/abiotic stress and grain quality	TASSEL	Bandillo et al. [35]
	12-way, funnel	Cultivars	1600 S9	Plant height and heading date	TASSEL	Li et al. [96]
	8-way, diallel	Breeding lines	1688 S5	Yield, plant height and heading date	TASSEL	Meng et al. [97]
	8-way, diallel	Cultivars	981 F6	Grain shape	GWAS	Ogawa et al. [82]
	4-way, diallel	Inbred lines	247 F7	Heading date	GWAS	Han et al. [23]
Maize	8-way, diallel	Inbred lines	1636 F6	Pollen shed, grain yield, and plant and ear height	QTLRel	Dell’Acqua et al. [32]
	4-way, funnel	Inbred lines	1291 F4:5	Plant height, ear height, and flowering time	GAPIT	Anderson et al. [89]
Barley	8-way, funnel	Old landraces and a model cultivar	5000 DH	Flowering time	mpMap	Sannemann et al. [60]
	32-way, funnel	Cultivars	324 F6	Climate and site-related agronomic adaptation	–	Bülow et al. [98]
Sorghum	29-way, diallel	Cultivars	~1000 S7	Plant height	TASSEL	Ongom and Ejeta [40]
Oats	8-way, diallel	–	600 S6	–	–	Aberystwyth University (unpublished)
*Legumes*						
Chickpea	8-way, diallel	Cultivars and breeding lines	~1200 F6	Heat tolerance	–	Gaur et al. [99]
Faba bean	11-way, open pollination	Inbred lines	>400 S9	Frost tolerance	–	Sallam and Martsch [54]
	4-way, funnel	Inbred lines	~1000 F4	Flower color and stipule spot pigmentation	–	Khazaei et al. [100]
Pigeonpea	8-way, diallel	Landraces and breeding lines	in progress	Resistance genes, maturing, and photoperiod	–	Saxena and Varshney [101]
Cowpea	8-way, diallel	Landraces and breeding lines	305 F8:10	Flowering, plant growth, seed size, and maturity	mpMap	Huynh et al. [52]
Soybean	8-way, funnel	Cultivars and exotic collections	764 F2:8	Yield under changing climatic conditions	–	Shivakumar et al. [102]
Groundnut	8-way, diallel	Breeding lines	~3000 F6	Seed traits	–	Pandey et al. [31]
	8-way, diallel	Breeding lines	in progress	*Aspergillus* resistance and aflatoxin contamination	–	ICRISAT (unpublished)
	8-way, diallel	Breeding lines	in progress	Drought tolerance	–	ICRISAT (unpublished)
	–	Breeding and commercial lines	in progress	–	–	Tifton, Georgia, USA
*Vegetables and fruits*						
Tomato	8-way, funnel	Cultivars and wild accessions	397 S3	Fruit weight	mpMap	Pascual et al. [37]
	8-way, funnel	Cultivars and wild accessions	400 F10	Resistance genes and fruit shape	–	Campanelli et al. [70]
	8-way, funnel	Cultivars and wild accessions	in progress	Morphoagronomic traits and resistance genes	–	Universitat Politècnica de València (unpublished)
Strawberry	6-way, diallel	Cultivars	1060 inter-cross	Fruit quality	–	Wada et al. [103]
Eggplant	8-way, funnel	Cultivars and wild accessions	in progress	Fruit traits	–	Universitat Politècnica de València (unpublished)
Pepper	8-way, funnel	Landraces	in progress	Fruit traits	–	Universitat Politècnica de València (unpublished)
*Industrial and oil crops*						
Cotton	12-way, funnel	Cultivars	1500 F7	Fiber yield and resistance genes	–	Li et al. [67]
	11-way, diallel	Cultivars and a breeding line	>550 S6	Fiber quality	TASSEL, GAPIT	Islam et al. [85]
Rapeseed	8-way	Elite cultivars	680 F6	Disease resistance, yield, plant architecture	–	Zhao et al. [104]
Chinese mustard	8-way, diallel	Breeding lines	408 F6	Quality traits (glucosinolate)	TASSEL	Yan et al. [87]

## 6. An Appraisal of MAGIC Populations Developed and Evaluated

Despite the complexity and resources required in developing MAGIC populations, their number is continuously growing, and there are a number of MAGIC populations already available (Table 2). Currently, new ones are in progress—in model species, as well as in economically important crops, including cereals, legumes, and vegetables (Table 2). Each MAGIC population has been established with different purposes, showing clear differences in population designs features as well as in the way that they are assessed. In the end, all of them have been used in different studies as breeding materials sources, demonstrating their capability to identify QTLs, thus strongly reducing the list of candidate genes related to complex agronomic traits. 

### 6.1. Model Species

The first set of MAGIC lines was obtained in the model species *Arabidopsis thaliana* (Table 2). The 19-way MAGIC population was developed by Kover et al. [51], and it served as a model and gave way to MAGIC development in other crops. With a final population size of at least 1026 S6 lines, it was used to identify candidate genes for germination date and bolting time. In this way, the authors identified two QTLs for germination date and four QTLs for bolting time. In the case of bolting time, the four QTLs explained 63% of the total phenotypic variance, and they seemed to be linked to well-known genes that affect flowering time. Some years later, Huang et al. [94] developed another eight-way MAGIC population to study flowering time and leaf morphology.

### 6.2. Cereals

The greatest number of MAGIC populations have been developed in cereals including wheat (nine), rice (eight), maize (two), barley (two), sorghum (one), and oats (in progress). Particularly, in wheat, there are different available MAGIC populations in both spring and winter wheat. Huang et al. [33] developed the first two populations, one eight-way MAGIC and, previously, another four-way MAGIC population as an intermediate stage between bi-parental populations and the eight-parent population. Just like some novel MAGIC populations, the final RILs were used to identify QTL associations with traits of interest like plant height [83,86]. A few years later, two different MAGIC populations were developed with about 1000–1500 F6:7 individuals to look for candidate genes involved in the control of awning and to assess QTLs for shoot length and coleoptile characteristics, respectively [38,92]. More recently, Stadlmeier et al. [71] constructed a reduced eight-way funnel MAGIC with 516 F6:8 RILs aimed at identifying QTLs for powdery mildew resistance.

In order to collect the broadest genotypic diversity present in rice, Bandillo et al. [35] not only developed eight-way MAGIC populations—one in *indica* and another one in *japonica* subspecies—but also inter-crossed the *indica* and *japonica* base populations to increase the overall diversity; this is referred to as the “global MAGIC” population. With the development of these MAGIC populations, they aimed to detect major QTLs related to disease resistance and tolerance to abiotic stresses transmitted from the founder lines, such as blast resistance, bacterial blight resistance, salt tolerance, and submergence tolerance. They also planned to evaluate the final RILs in field trials to identify lines adapted to a range of production constraints in the major productor countries of Asia and Africa, particularly stress tolerance. Novel 12-way, eight-way, and four-way MAGIC populations aimed to identify QTLs related to heading dates [23,96,97]. 

Two MAGIC populations have been developed in barley—eight-way and a 32-way ones with population sizes of 5000 DH and 324 F6, respectively [60,98]. These populations have been used to study flowering time and climate and site-related agronomic adaptations. In the case of maize and sorghum, a study of the 1000–1500 final RILs was focused in traits related to plant architecture like plant height [32,40,89]. QTLs and candidate genes were suggested for two important and complex traits such as grain yield and flowering time [32]. 

### 6.3. Legumes 

Many of the legume MAGIC populations have already been developed or are still in progress at International Crops Research Institute for the Semi-Arid Tropics (ICRISAT) including those of chickpea, pigeonpea, and groundnut [105]. The highly recombinant chickpea MAGIC population with approximately 1200 F6 RILs allowed for the development of several heat-tolerant progenies, while the aim of the still in progress pigeonpea MAGIC is to look for resistance genes and early maturing and photoperiod intensive lines [101,106]. Three additional groundnut eight-way MAGIC populations targeting different trait combinations are currently under development [31,69,107,108]. While the first one focused on seed traits like fresh seed dormancy or oil content, the second MAGIC population was developed with *Aspergillus* resistance and aflatoxin contamination as targets, and the third one tries to dissect components of drought tolerance.

Two MAGIC populations have been developed in the faba bean. The 11-way faba bean MAGIC with less than 400 S9 final RILs aimed to find QTLs related to frost tolerance, while the four-way one with approximately 1000 F4 was focused on flower color and stipule spot pigmentation [54,100]. There are also cowpea and soybean MAGIC populations, both focused on agronomic traits related to yield potential under changing climatic conditions [52,102]. 

### 6.4. Fruits and Vegetables 

Some MAGIC populations have already been developed in the tomato (two) and the strawberry (one). However, to our knowledge, there are others in progress in eggplants and peppers, as well as a new one in tomatoes, which is in the last stages of development at Universitat Politècnica de València (Spain). The main goal of these populations is to dissect QTLs related to agronomic and fruit quality traits.

To widen the genetic diversity of the tomato, Pascual et al. [37] developed the first tomato MAGIC population, selecting four accessions of cultivated tomato (*Solanum lycopersicum*) and four of weedy tomato (*S. lycopersicum* var. *cerasiforme*) as founders, including cherry tomato accessions, to study fruit weight. More recently, another tomato MAGIC population was developed by Campanelli et al. [70] with the aim of developing organic tomato genotypes by participatory plant breeding (PPB). In this case, they constructed the first reported inter-specific MAGIC population by inter-crossing seven *S. lycopersicum* accessions and one wild accession of *S. cheesmaniae*. MAGIC lines were cultivated in different organic farms with different locations to carry out the PPB, thus showing significant phenotypic differences in development, productivity, and fruit color. This variability was used to select families of tomatoes adapted to low input crop management, different environments, agricultural practices, and market conditions to be directly released as new varieties. 

The six-way strawberry MAGIC population was used to study the major agronomic fruit traits, including flowering habit, fruit weight, fruit skin color, and fruit firmness, as well as quality fruit traits such as soluble solid content or titratable acid [103]. 

### 6.5. Industrial and Oil Crops

Li et al. [67] developed an upland cotton MAGIC population with a total of 1500 F7 final RILs by inter-crossing 12 founder lines with the aim of increasing the intra-variety genetic diversity based on existing germplasm resources and improving fiber yield and quality. With the same objective, Islam et al. [85] developed another 11-way cotton MAGIC population. In the case of oil seed crops, a rapeseed MAGIC population was developed with 680 F6 final RILs to study disease resistance, yield potential, plant architecture, and ecological adaptability [104]. Recently, a Chinese mustard (*Brassica juncea*) MAGIC population was developed with a final population size of 408 F6 RILs mainly focused on glucosinolate-related traits to improve fruit quality [87]. Relevant QTLs were used to predict candidate genes associated with glucosinolate synthesis. 

## 7. Future Prospects

Currently available MAGIC populations including high phenotypic and genetic diversities have demonstrated their potential for QTL fine mapping. However, there is still a wide range of possibilities to be exploited. 

### 7.1. Inter-Specific MAGIC Populations

It has been demonstrated in many crops that there has usually been a significant loss of genetic variability of the cultivated species because of domestication processes and plant genetic breeding programs [109,110]. This lack of genetic variation implies a higher risk of production losses and reduced yields when facing threats derived from different types of stress, and it also limits the exploitation to intraspecific variation [111], thus constraining the sources of variability for crop breeding programs. In this regard, with the aim of recovering part of this lost variability, geneticists and plant breeders have fallen back on related species and wild relatives as sources to broaden the genetic base of cultivated species [112]. Including these wild species in multi-parent population schemes to get progenies from distant crosses opens new avenues for the exploitation of this variation.

MAGIC populations could be used as a way of including multiple genomic fragments or introgressions of one or several wild species into a cultivated background genome. Introducing a single wild relative among the eight founders of an inter-specific MAGIC population will include approximately 12.5% of the wild genome in the final RILs, similarly to a BC2 bi-parental population. This approach not only allows for the study of foreign QTLs in a mixed background of the eight parents but also for the development of genetically characterized elite materials that can be easily incorporated in breeding programs. The novel eggplant and tomato populations that are still in progress at Universitat Politècnica de València are inter-specific MAGIC populations including wild accessions as founders. The aim here is to widen the genetic base of the crop and to take profit advantage of wild donor introgressions for relevant traits, including adaptation to climate change.

It is important to consider that on a practical level, inter-specific populations could present some drawbacks, e.g., crossing barriers, low hybrid fertility, sterility, or undesirable agronomic traits. In addition, linkage drag due to reduced recombination rate at introgressed fragments could be observed, and homozygosity could not be reached in some genomic regions due to detrimental effects, resulting in negative selection [112]. Some of these issues have been observed in the novel eggplant MAGIC population, in which an accession of the wild species *Solanum incanum* has been used as a female founder. This inter-specific crossing drags the maternal cytoplasmic background of the wild parent in some of the lines, and, on average, this reduces fertility, as has been observed in alloplasmic eggplant materials [113]. It has been observed that these lines present a greater difficulty in fruit setting, as well as in the production of viable seeds, which suggests a lower viability of pollen. However, it also allows for the evaluation of the interactions of a wild cytoplasm with a mostly cultivated nuclear genome, thus opening new ways for the understanding of the effects of cytoplasm on the phenotype of important traits.

### 7.2. MAGIC-Like Approximations 

New strategies and designs could be developed to generate multiparent resources for plant species for which founder pure lines cannot be obtained or may take too much time to obtain. In this way, multi-parental populations have been already developed in some fruit trees like apple and strawberry, confirming the expected advantages of multi-population studies [103,114]. The main problem of these species with QTL identification is the high heterozygosity and the appearance of false-positives due to relationships between individuals. In these cases, it is important to follow the pedigree to understand the kinship relationships for their analysis. In addition, considering marker data for common ancestors makes it possible to trace the source of favorable alleles.

Vegetative (or clonal or asexual) propagated populations consist of highly heterozygous clones that are genetically identical to their parents and can be conserved and utilized by continued vegetative reproduction for a long period of time [115]. Most clonal species have the problem of inbreeding depression, but hybridization between different clones, or even the self-pollination of one clonal line, can produce seeds and therefore generate segregating clonal F1 progenies [116]. Taking a four-way MAGIC population as an example with four inbred lines A, B, C, and D as parents, the hybrid made between inbred lines A and B will be equivalent to the female parent of a clonal F1 population after the female haploid building, and the hybrid made between inbred lines C and D will be equivalent to the male parent of a clonal F1 population after the male haploid building [116]. Then, a double cross (or four-way cross) will be made between the two hybrids—one used as female and the other one as male. Subsequent cycles of hybridization between the clones can increase the admixing of the genomes and recombination, thus producing a pseudo-MAGIC population of clones that can be vegetatively propagated.

We also suggest the possibility of developing MAGIC-like populations in crops where selfing cannot be applied due to self-incompatibility, which makes pure lines impossible to obtain. This is a case similar to the so-called heterogeneous stocks and Collaborative Cross populations in mice [80]. In this case, as occurred for the generation of the mice CC lines, repeated generations of inbreeding through sibling mating can be applied to increase the degree of fixation.

### 7.3. Incorporation of MAGIC Populations in Breeding Pipelines 

Apart from the inherent value of MAGIC populations as experimental materials as mapping populations and for the detection of genes and QTLs, they may also represent an elite material for breeders [22,52]. Given the nature of MAGIC populations, in which parents generally are complementary for traits of interest [15,24], new phenotypes will arise in the MAGIC population, and it may be possible to recover in the final MAGIC population lines that display improved characteristics that can be used as elite material for breeding programs or even directly as new cultivars developed through selection within MAGIC populations [99,117]. In this respect, in a MAGIC population perfectly fixed in homozygosis, the frequency of specific genotypes in the final MAGIC population (as long as genes are unlinked) can be calculated as the product of the frequency of the desired alleles of each target locus among the founders. For example, the expected frequency of a genotype in a MAGIC population with eight founders in which the desired genotype corresponds to four unlinked genes with frequencies of 1/2, 1/4, 1/8, and 3/8 among the founders would be 3/512, and the minimum size for getting one individual with this genotype with a probability of 0.95 would be of 510 individuals. Therefore, given the large population sizes, it may be possible to find desired combinations of pyramided genes or QTLs among the MAGIC lines, even for multiple target loci, that may be of interest to be directly incorporated into the breeding pipelines by breeders. In this respect, as proposed by Scott et al. [24], the availability of MAGIC population ‘packages,’ which integrate the MAGIC population material plus extensive phenotypic and genotypic characterization data, would greatly facilitate the incorporation of MAGIC materials in breeding pipelines.

MAGIC lines can be of particular interest as parents of hybrids. In this way, the evaluation of specific and general combining abilities [118] by crossing MAGIC lines with testers may result in either the identification of MAGIC lines that give heterotic hybrids with already established elite lines or even in the establishment of heterotic groups within MAGIC populations. The availability of genotyping data in the MAGIC populations can also facilitate the use of genetic distances, or other genetic parameters, among lines as parameters to predict performance of hybrids [119]. 

We also propose that MAGIC populations can also be used, mimicking the composite crosses evolutionary breeding strategy [120,121], to let natural selection act in highly diverse MAGIC populations—either after the final population has been obtained or during their development. Composite crosses that have been let evolve under cultivation conditions have proved valid ways to increase yields in cereals by exploiting adaptation to local environments [122,123]. Though intentional selection is often avoided during the development of MAGIC populations [17], artificial selection may contribute to increasing the frequencies of alleles of interest for specific loci, facilitating the incorporation of MAGIC materials in breeding programs. In this respect, Campanelli et al. [70] developed a tomato MAGIC population in which participatory breeding was applied during the developmental phases of the MAGIC population. Additionally, given the interest in heterogeneous materials for organic agriculture [124], subsets of MAGIC populations, or even the whole population, may be directly used for cultivation to obtain a highly diverse produce of interest for specific markets demanding diversity.

Finally, as has been proposed for experimental populations with introgressions from crop wild relatives [112], the establishment of public–private partnerships (PPPs) for the development of MAGIC populations may greatly stimulate the use of MAGIC populations in breeding pipelines. In this way, developing MAGIC populations requires a long time, particularly for crop plants where only one or a few reproductive cycles can be achieved per year and a significant amount of resources are needed for their development [37,38], and PPPs may optimize the resources and expertise for the development of MAGIC populations; at the same time, breeders from private companies can spot interesting genotypes that can be followed in a pedigree. Several examples of successful PPPs in breeding exist [125,126,127], and the nature of MAGIC populations, where a great diversity is usually present in founders, as well as the many new phenotypes and genetic combinations that arise during the development of the MAGIC population, is of particular interest to breeders and may greatly spur the use of MAGIC materials in breeding for the development of new cultivars. 

## 8. Conclusions

MAGIC populations are recombinant inbred sets obtained after intercrossing multiple parents. Although the development of MAGIC populations require considerable efforts, theoretical and real studies show that MAGIC populations represent powerful tools for the detection of QTLs present in the set of parents, with considerable advantages over bi-parental and germplasm sets for the detection of QTLs. An important feature of MAGIC populations is that recombinant elite lines can be directly selected by breeders for being introduced in breeding programmes, or directly selected as new varieties. As the number of MAGIC populations is continuously growing, new tools for an efficient analysis of MAGIC populations have recently been developed. Also, new developments can extend the MAGIC approach to crops in which development of standard MAGIC populations are difficult to be obtained. We are confident that MAGIC populations will play an important role in addressing the formidable challenges faced by breeders in a scenario of climate change and the increased demand of plant products [3,4,5] by significantly contributing to the development of new generations of resilient, highly productive, and resource-efficient cultivars.

## Figures and Tables

**Figure 1 biology-09-00229-f001:**
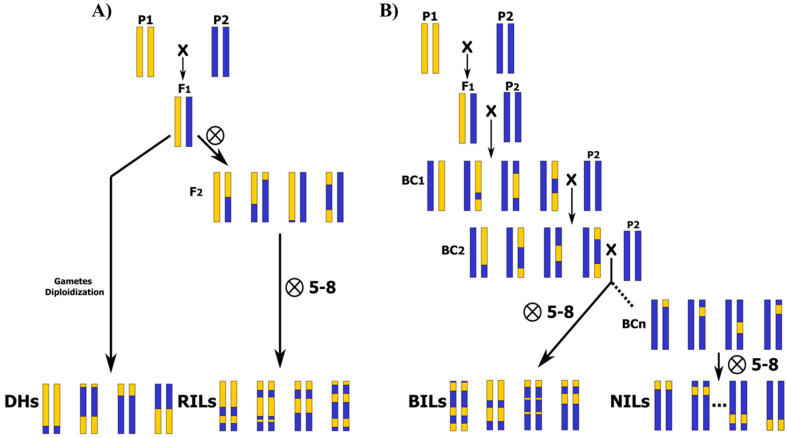
Bi-parental populations, from lowest to highest number of recombination events and required efforts in population development. Parental founders P1 and P2 are contrasting for the trait/s of interest, and F1 is the simple hybrid derived from founder cross: (**A**) Doubled haploids (DHs), F2, and recombinant inbred lines (RILs) developed by successive generations of selfing. (**B**) Backcross (BC), backcross inbred lines (BILs), and near isogenic lines (NILs) obtained by successive generations of selfing.

**Figure 2 biology-09-00229-f002:**
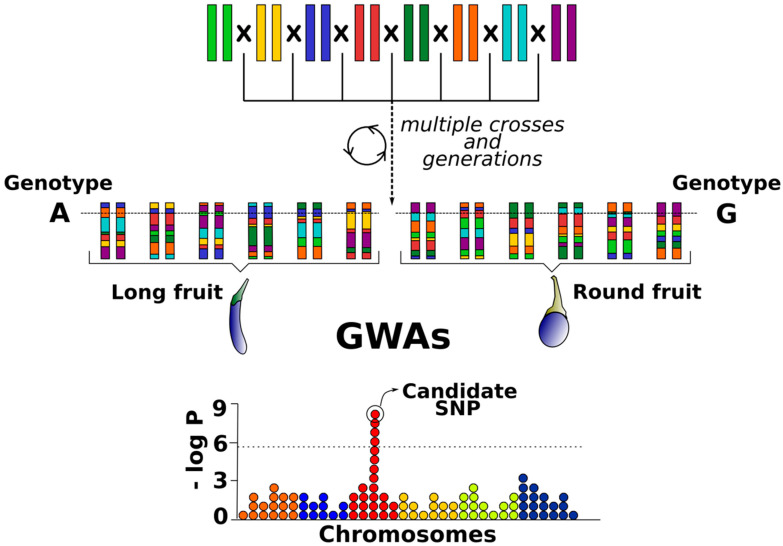
Historical recombination events and natural genetic diversity of germplasm collections used for genome-wide association studies (GWAS).

**Figure 3 biology-09-00229-f003:**
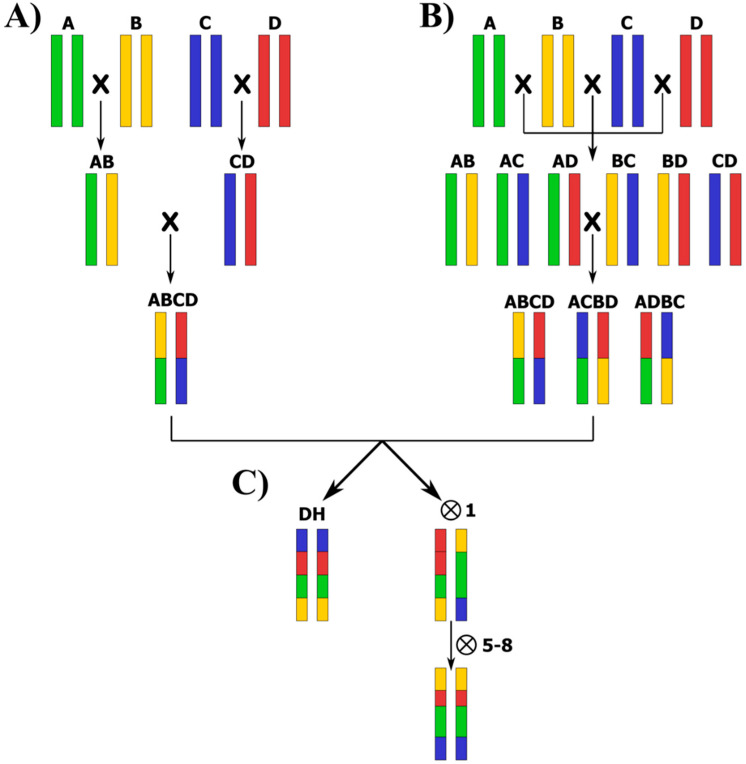
Cross-designs of a 4-way MAGIC population where the founders are A, B, C, and D: (**A**) “funnel” design; (**B**) “diallel” design; and (**C**) achievement of homozygous individuals by doubled haploids (DH) production, or by several rounds of selfing following the single-seed descent (SSD) method.

**Figure 4 biology-09-00229-f004:**
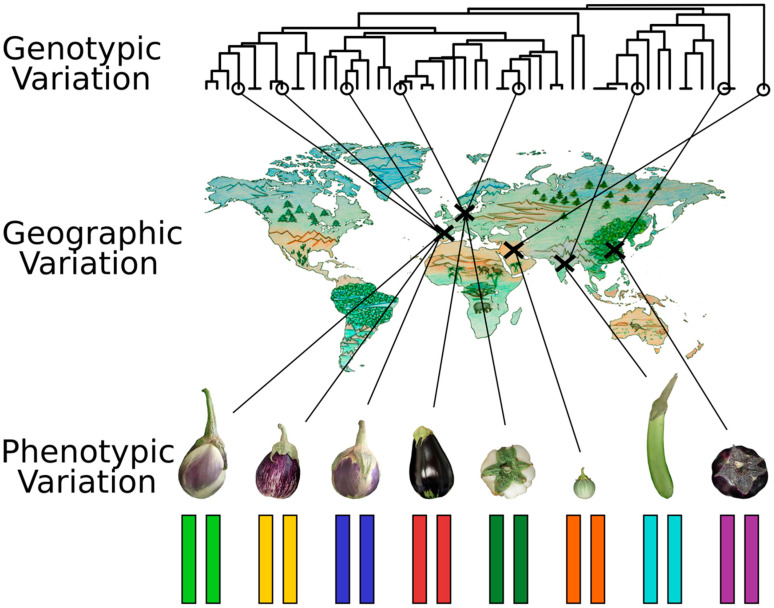
Maximizing founder selection based on phenotypic, genetic, and geographic diversity when using eggplant as a model.

**Figure 5 biology-09-00229-f005:**
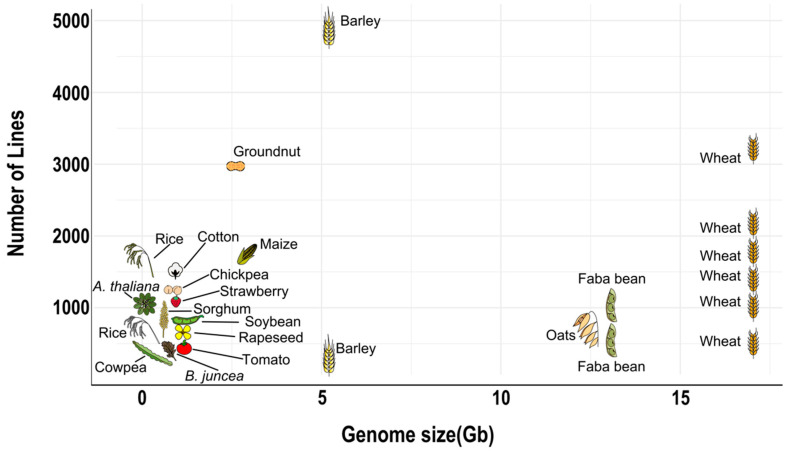
Population size of MAGIC populations developed compared to their genome size. Detailed information in Table 2.

**Figure 6 biology-09-00229-f006:**
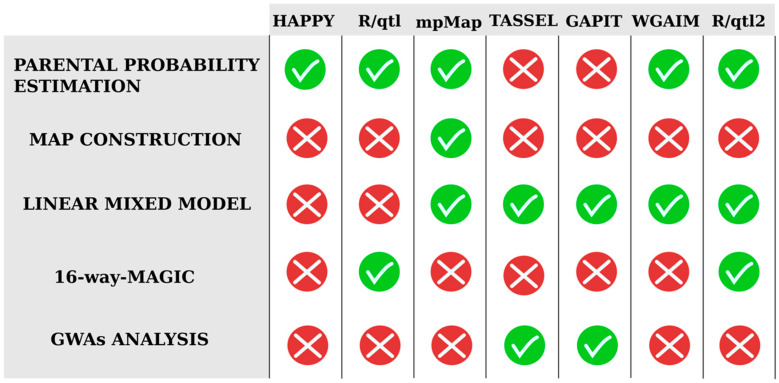
Analysis software used for QTL analysis in the available MAGIC populations, indicating the type of functionalities and analyses that can be performed in each of them.

**Table 1 biology-09-00229-t001:** Advantages (+) and limitations (-) of linkage analysis (bi-parental populations), association mapping (GWAS) using germplasm populations, and multi-parent advanced generation inter-cross (MAGIC) for different characteristics of interest for their development and use. QTL: quantitative trait locus.

Characteristic	Population Type		
	Bi-Parental	Germplasm	MAGIC
Investment in time to be established	-	+	- -
Required population size	+	-	-
Genetic and phenotypic diversity	-	+ +	+
Suitability for coarse mapping	+	-	+
Suitability for fine mapping	-	+	+
QTL resolution	-	+	+ +
Required marker density	+	-	-
Recombination rate	-	+ +	+
Low population sub-structure	+	-	+
Low LD	+	-	+

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
