# Peer review of "The Dawn of the Age of Multi-Parent MAGIC Populations in Plant Breeding: Novel Powerful Next-Generation Resources for Genetic Analysis and Selection of Recombinant Elite Material"

_biology, 2020, doi:10.3390/biology9080229_

Round 1

Reviewer 1 Report

The review by Arrones et al., focus on the potential of MAGIC populations for plant breeding, including comprehensive information on the different types of mapping populations for QTLs detection, and the advantages and limitations of MAGIC populations, as well as strategies for development, and software available for their genetic analysis.

The manuscript is really well written clear and informative in all its sections. Figure are well done.

Minor points:

Line 55 small English conjugation problem

Line 87 The title material and methods in a review is quite misleading. I suggest finding a more appropriate section title, such as “overview on different population type” or similar or simply remove it.

Line 580-581: is reported “This section is not mandatory but may be added if there are patents resulting from the work reported in this manuscript.” Is it a wanted sentence, or is it a comment leftover of manuscript writing?

Line 645 small English grammar problem

Author Response

The review by Arrones et al., focus on the potential of MAGIC populations for plant breeding, including comprehensive information on the different types of mapping populations for QTLs detection, and the advantages and limitations of MAGIC populations, as well as strategies for development, and software available for their genetic analysis.

The manuscript is really well written clear and informative in all its sections. Figure are well done.

 - We thank the reviewer to highlight the interest and usefulness of our review and for the constructive comments that have enhanced the quality of the manuscript. The modifications made in the manuscript in response to specific comments are detailed below.

Minor points:

Line 55 small English conjugation problem

- According to the suggestion of the reviewer, we have rephrased the sentences as follows (lines 53-58): "The success of conventional breeding was enhanced by genetic transformation technologies that were applied since the 1980s, allowing to achieve important advances in plant breeding [3]. Nevertheless, many of those technologies have been mostly focused on monogenic traits while many major agronomic traits of interest are quantitative, controlled by multiple loci and generally with a large environmental influence [6–9], posing a challenge to breeders due to the limited efficiency of breeding based on phenotypic selection.".

Line 87 The title material and methods in a review is quite misleading. I suggest finding a more appropriate section title, such as “overview on different population type” or similar or simply remove it.

- According to the suggestion of the reviewer, we have changed the title of section 2 from "Materials and Methods" to "Overview of experimental populations and germplasm collection for traits dissection".

Line 580-581: is reported “This section is not mandatory but may be added if there are patents resulting from the work reported in this manuscript.” Is it a wanted sentence, or is it a comment leftover of manuscript writing?

- According to the suggestion of the reviewer, we have removed the sentence.

Line 645 small English grammar problem

- According to the suggestion of the reviewer, have rephrased the sentence as follows (lines 642-643): "We also suggest the possibility of developing MAGIC-like populations in crops where selfing cannot be applied due to self-incompatibility, thus pure lines cannot be obtained".

Reviewer 2 Report

The work presented by Arrones at al is relevant and reviews an important thematic, that falls within the scope of the journal. The authors approach the relevance of MAGIC tool in terms of climate change, making the subject of broad interest and timely. Figures have quality and are informative.

I don't find the (+) and (-) very explicative in Table 1. I think the authors could replace the symbols with a word or 2 that could describe the advantages/disadvantages better.

L55: delete 'were'

L77: replace 'dramatic' with a different word

L136: 'may be' not 'maybe'

L254: delete '1'

Author Response

The work presented by Arrones at al is relevant and reviews an important thematic, that falls within the scope of the journal. The authors approach the relevance of MAGIC tool in terms of climate change, making the subject of broad interest and timely. Figures have quality and are informative.

 - We thank the reviewer to highlight the interest and relevance of our review and for the constructive comments that have enhanced the quality of the manuscript. The modifications made in the manuscript in response to specific comments are detailed below.

I don't find the (+) and (-) very explicative in Table 1. I think the authors could replace the symbols with a word or 2 that could describe the advantages/disadvantages better.

- We thank the reviewer for the suggestion, however, our idea for Table 1 was to summarize, in a visual and intuitive way, the advantages and limitations of each population type. In fact, in the body of the manuscript, all the characteristics for each population are extensively detailed and explained, thus we do not consider that it is necessary a further description of the characteristics in Table 1.

L55: delete 'were'

- We have made the change proposed by the reviewer.

L77: replace 'dramatic' with a different word

- We have replaced “dramatic” with “remarkable”

L136: 'may be' not 'maybe'

- We have made the change proposed by the reviewer.

L254: delete '1'

- We have made the change proposed by the reviewer.